# Impact of Environmental Factors on the Prevalence Changes of Allergic Diseases in Elementary School Students in Ulsan, Korea: A Longitudinal Study

**DOI:** 10.3390/ijerph17238831

**Published:** 2020-11-27

**Authors:** Jiho Lee, Seokhyun Yun, Inbo Oh, Min-ho Kim, Yangho Kim

**Affiliations:** 1Department of Occupational and Environmental Medicine, Ulsan University Hospital, University of Ulsan College of Medicine, Ulsan 44033, Korea; kkami2000@hanmail.net; 2Environmental Health Center, University of Ulsan College of Medicine, Ulsan 44033, Korea; oinbo@naver.com; 3Department of Informatics, Ewha Womans University Seoul Hospital, Seoul 07804, Korea; mino-kim@naver.com

**Keywords:** allergic diseases, prevalence, environmental factors, longitudinal

## Abstract

We examined the effect of long-term changes in environmental factors on the prevalence of allergic diseases in elementary school students in Ulsan, Korea. This longitudinal study was conducted among 390 students who were enrolled from three elementary schools in Ulsan in both the first (2009–2010) and second survey (2013–2014). The International Study of Asthma and Allergies in Childhood questionnaire was used to determine the prevalence of allergic diseases and hazardous environmental factors. Generalized estimating equations (GEE) were used to investigate the impact of environmental factors on the change in the prevalence of allergic diseases. The survey revealed that an increased risk of allergic rhinitis was associated with exposure to secondhand smoke, the remodeling of a room, the replacement of wallpaper or flooring, the use of a humidifier, and irritation symptoms of air pollution. Atopic dermatitis was associated with the relocation to or repairs of a new house, and allergic conjunctivitis was associated with low levels of weekly physical activity, the use of insecticides, and irritation symptoms of air pollution. The results indicate that (1) allergic rhinitis and atopic dermatitis are associated with indoor pollution, (2) allergic conjunctivitis is associated with exposure to indoor chemical compounds and low levels of weekly physical activity. This study suggested that the proper evaluation and decrease in the number of environmental risk factors could effectively manage allergic diseases.

## 1. Introduction

The prevalence of childhood allergic diseases varies widely throughout the world. The factors causing this variation in prevalence differ from one location to another and from one age group to another. Generally, these factors are related to lifestyle, dietary habits, microbial exposure, economic status, an indoor or outdoor environment, climatic variation, and the awareness of the disease or the management of symptoms [1]. In Korea, the prevalence of allergic diseases has been increasing steadily since 1990, thereby decreasing patients’ quality of life and severely increasing their socioeconomic burden [2].

The increasing prevalence of allergic diseases in developed countries indicates that environmental risk factors and lifestyle, rather than genetic predisposition, are the major determinants of allergic disease.

The factors associated with the prevalence of allergic diseases and comorbidity are varied. Climate-related factors, including ambient temperature and outdoor humidity, are associated with allergic diseases symptoms [3]. Dietary factors and socioeconomic status may also affect the prevalence directly and indirectly. Air pollution not only aggravates the symptoms of allergic diseases but also causes their onset in new individuals. In specific, oxidizing agents, including nitrogen oxides and ozone, are well recognized as important allergy- associated pollutants [4].

In addition, allergic diseases appear to progress differently throughout life depending on the environmental risk factors present in the child’s developmental period [5]. For example, the condition of indoor air is worsened by activities such as the use of home appliances, detergents, cosmetics, and fuel for cooking or heating, as well as by the surfaces of furniture and the building itself [6]. Spending a lot of time at home, the prevalence of elementary school students are greatly affected by indoor air pollutants, even without any previous allergic disease during their growth period [7].

Numerous epidemiological studies aiming to identify the relationships between allergic diseases and environmental factors have been conducted, most of which were cross-sectional studies. Individual follow-up studies in Korea are rare. Hence, this study, which was longitudinal in design, aimed to evaluate the impact of environmental risk factors on the changes in the prevalence of different allergic diseases in a cohort of elementary school students in Ulsan Metropolitan City.

## 2. Materials and methods

### 2.1. Study Subjects

The first survey was conducted for students in grades 1 and 2, aged 6 to 9 years old, across 3 elementary schools between 2009 and 2010. A second survey was then conducted for the same subjects between 2013 and 2014 in Ulsan City. A total of 392 elementary school students were enrolled in both surveys, and the questionnaire was completed by their parents within 1 week of enrolment.

The standardized International Study of Asthma and Allergies in Childhood (ISAAC) Phase I written core questionnaire, which was translated into Korean, was used to elicit information from the parents. Having been used in some previous studies carried out in Korea [2,8,9], this questionnaire is a well-known and validated method in Korean studies.

An additional questionnaire was used to obtain information on demographic characteristics and potential risk factors for allergic disease including sex, family history of allergy, the presence of a physician who diagnosed and/or a physician who treated the allergic disease, and socioeconomic status (income). This questionnaire was also used to collect indoor/outdoor environmental information, such as secondhand smoking at home, daily ventilation time, pet ownership, the use of home appliances (sleeping beds, humidifiers, or air conditioners), the use of insecticides, instances of house repairs (windows, wallpaper, or floors), moving into a new house, and the daily activities of the subject (time spent in front of the television and computer, time spent on physical activity).

The prevalence of allergic disease in this study was calculated based on answers to the survey questions: “Have you been diagnosed with allergic diseases (asthma, allergic rhinitis, atopic dermatitis or allergic conjunctivitis) by a physician in the last 12 months”? and “Have you been treated for allergies in the last 12 months”? Since asthma was only present in 2 subjects, these subjects were excluded from the analysis, giving a total of 390 subjects.

### 2.2. Statistical Analysis

Operational definitions were used in this study to categorize the changes in the prevalence and risk factors for allergic disease between the first and second survey. If the disease was not present in the first investigation but was in the second, the subject’s disease state was defined as “worsened (incident case)”. In the opposite case, the subject’s disease state was defined as “improved”.

The McNemar test was used to investigate the associations between changes in the prevalence of allergic disease (physician diagnosed or treated) and environmental factors during the follow-up period, while the McNemar test or paired t test were performed to investigate differences in categorical and continuous variables.

Generalized estimating equations (GEE) were used to evaluate the effect of the changes in environmental risk factors on the prevalence of different allergic diseases. The calculated odds ratio (OR) was obtained after adjusting for age, sex, and family history. The statistical analyses were carried out by Statistical Analysis System (SAS, 9.4 SAS Institute Inc., Cary, NC, USA).

### 2.3. Ethical Consideration

The study was approved by the Institutional Review Board of Ulsan University Hospital (IRB no. UUH 2009-09-061).

## 3. Results

The general characteristics of the subjects at the time of the first survey are shown in Table 1. The mean age of participants was 8.4 years. The average follow-up period of the study subjects was 49 months (range, 42–58 months). The prevalence of physician-diagnosed allergic rhinitis, atopic dermatitis, and allergic conjunctivitis was 88 (22.6%), 37 (9.5%), and 37 (9.5%), respectively. A maternal history of allergic rhinitis, atopic dermatitis, and allergic conjunctivitis was 86 (22.3%), 14 (3.7%), and 28 (7.4%), respectively (Table 1).

Results of the McNemar test indicating changes in allergic disease prevalence between the first and second survey, based on physician diagnosis and the treatment of allergic diseases (allergic rhinitis, atopic dermatitis, and allergic conjunctivitis), are shown in Table 2. Regarding cases of physician-diagnosed allergic rhinitis, 13.3% (50/376) of the subjects had newly developed disease (incident cases) in the second survey, while 9.8% (37/376) of the subjects had improved disease during this follow-up period. The percentage of incident cases of atopic dermatitis was 4.7% (18/381), while the percentage of improved cases was 6.8% (26/381). The percentage of incident cases of allergic conjunctivitis was 9.8% (37/378), while the percentage of improved cases was 6.3% (24/378).

With regard to cases of physician-treated allergic rhinitis, 13.2% (49/371) of subjects had newly developed disease (incident cases) in the second survey, while 12.7% (47/371) had improved disease during the follow-up period. The percentage of subjects with incident cases of atopic dermatitis was 3.8% (14/372), while the percentage of improved cases was 6.5% (24/372). The percentage of incident cases of allergic conjunctivitis was 10.2% (38/374), while the percentage of improved cases was 6.1% (23/374) (Table 2).

Table 3 shows the difference in environmental risk factors for allergic diseases (continuous variables) between the first and second survey. Household income, daily time spent in front of the television and computer, daily time spent at school, and age of the current household building increased significantly between the first and second survey. However, the time spent participating in moderate weekly physical activity, time spent at home, and daily ventilation time significantly decreased.

For the categorical variables, the results showing the changes in individual environmental risk factors for allergic diseases between the first and second survey are shown in Table 4. The frequency of the presence of several environmental factors increased significantly over the follow-up period. These significant factors were the presence of paternal smoking, the remodeling of a room, the replacement of wallpaper or flooring, the use of a children’s sleeping bed, the use of a humidifier, the use of an air conditioner, the use of insecticides, the relocation to or repair of a new house, and pet ownership.

Results of the GEE used to determine the effects of environmental risk factors on allergic disease prevalence are shown in Table 5. The environmental risk factors that significantly influenced the diagnostic prevalence of allergic rhinitis included a maternal history of allergic rhinitis, exposure to secondhand smoke, the use of a humidifier, and irritation symptoms from air pollution. The factors that significantly affected the treatment of allergic rhinitis were the remodeling of a room, the replacement of wallpaper or flooring, a maternal history of allergic rhinitis, exposure to secondhand smoking, and the use of a humidifier. We found that these environmental factors increased the risk of allergic rhinitis, after controlling for genetic inheritance.

In terms of the diagnostic prevalence of atopic dermatitis, a maternal history of atopic dermatitis and the relocation to or repair of a new house significantly increased the prevalence of atopic dermatitis. However, it was found that a history of atopic dermatitis in siblings and time spent at school significantly decreased the prevalence of the disease. Similarly, a maternal history of atopic dermatitis and the relocation to or repair of a new house significantly increased the therapeutic prevalence of atopic dermatitis, while a history of atopic dermatitis in siblings significantly decreased it.

For allergic conjunctivitis, factors found to significantly increase diagnostic prevalence included a maternal history of allergic conjunctivitis, household income, moderate levels of physical activity per week, the use of insecticides, and irritation symptoms of air pollution. On the other hand, painting inside the house and the use of hair spray was associated with a significant reduction in diagnostic prevalence. With regard to therapeutic prevalence, a maternal history of allergic conjunctivitis, house income, the use of insecticides, and irritation symptoms of air pollution significantly increased disease risk. Meanwhile, the painting of an internal walls, contrary to expectations, was associated with a significantly reducing the risk.

## 4. Discussion

Although the ISAAC questionnaire has been used widely in Korea to evaluate the relationship between the prevalence of allergic diseases and associated factors [2,8,9], most of these studies were cross-sectional. This has made it difficult to determine the relationships between allergic diseases and environmental factors. Due to recall and selection bias, neither retrospective nor cross-sectional studies are suitable for the assessment of cause–effect relationships between the development of allergic diseases and the exposure to allergens and adjuvant factors [10]. In this study, in order to reduce bias, we calculated the prevalence of disease based on the diagnosis or treatment by a physician. We also investigated changes in the prevalence over a four-year period (42 to 58 months), while monitoring data to identify environmental risk factors for allergic diseases in a cohort of elementary school students.

Allergic disease develops in individuals with a genetic predisposition after they are exposed to environmental factors. A family history of atopy increases the risk of allergic disease in children, and this has been shown in long-term prospective studies [11,12]. Our study also confirmed that maternal or sibling history of atopy was a risk factor for allergic rhinitis, atopic dermatitis, and allergic conjunctivitis.

The incident cases of allergic rhinitis and allergic conjunctivitis increased more than those of atopic dermatitis (13.3%, 9.8%, and 4.7% respectively) during the follow-up period. Additionally, the changes in treatment-based prevalence showed similar patterns. These results are in agreement with those of previous studies, which indicated that each of these atopic disorders has a different peak age of onset and sequential progression of the disease [13,14].

The development of allergic diseases is determined not only by genetic predisposition but also by environmental and lifestyle factors [5,15,16]. In industrialized countries, most people with a “Western lifestyle” spend more than 95% of their time indoors; in particular, children are spending more time indoors than ever before. This study revealed that there were changes in the lifestyle of students over the study period. As school grade increased, the time spent per week participating in physical activities significantly decreased, while the time spent in front of the television and computer and at school increased during the study period.

Over the study period, the indoor environment of homes also changed; they became more suitable for dust mite population growth and air contamination. In addition, the number of indoor pets and the use of sleeping beds increased, resulting in lower air-exchange rates. Furthermore, significant changes in environmental factors, such as paternal smoking, the remodeling of a room or floor, the use of home appliances such as humidifiers and air conditioners, the use of insecticides, and the relocation to or repair of a new house, resulted in a substantial increase in the concentration of potentially harmful substances in the indoor air.

Recently, technology designed to reduce energy consumption has led to us making homes more airtight and heavily insulated. This has resulted in an increased likelihood of chemical compounds being retained indoors and, in turn, increased air contamination. Various chemical compounds are emitted into indoor spaces by a wide range of sources, including building and indoor materials, furniture, curtains, air fresheners, deodorants, pesticides, personal care products, cooking, and other everyday household products [10,17,18].

Based on the results of GEEs, the allergic diseases that were commonly affected by genetic predisposition were also those that were affected in various ways by environmental factors.

The prevalence of allergic rhinitis was affected by the presence of secondhand smoke, the remodeling of a room, the replacement of wallpaper or flooring, the use of a humidifier, and irritation symptoms of air pollution. It is thought that many indoor pollutants, such as volatile organic compounds, are generated when replacing wallpaper or flooring and when repairing a room or wall. These, therefore, may constitute significant risk factors for the development of allergic rhinitis. These environmental factors have been found to increase the risk of allergic rhinitis in environments with a low rate of ventilation in previous studies [16,19,20].

For atopic dermatitis, relocation to or repair of a new house increased both diagnostic and therapeutic prevalence. This could have been caused by building materials and adhesives, which are sources of volatile organic compound emissions, particularly in newly built houses [18,21]. However, a history of atopic dermatitis in siblings and more time spent at school significantly decreased the risk of atopic dermatitis. This contradictory result indicates that genetic predisposition differs between siblings. A possible explanation for this result is that, if one sibling has developed atopic dermatitis, the parents may take steps to manage the risk of the disease in the other children. In addition, family size (number of siblings) has been hypothesized to be inversely related to the risk factors for allergic disease [10,13].

For allergic conjunctivitis, household income (rank), moderate physical activity per week, the use of insecticides, and irritation symptoms of air pollution increased the diagnostic prevalence of disease. On the other hand, the painting of internal walls and the use of hair spray decreased the diagnostic prevalence of allergic conjunctivitis. Given the findings on allergic conjunctivitis, it is thought that the children included in this study were not sensitive to these risk factors known to aggravate allergic conjunctivitis.

The development and phenotypic expression of atopic disease depends on a complex interaction between genetic factors; environmental exposure to allergens; and non-specific adjuvant factors such as tobacco smoke, air pollution, the host’s microbiome, lifestyle, and infections [10,22]. Additionally, the hygiene hypothesis conceptualized that childhood exposure to microorganisms will protect also against allergy [3]. The exposome, which is the sum total of all the exposures of an individual in a lifetime, has undergone major shifts in the last few decades, affecting human health and disease [23]. Urbanization has led to a reduced biodiversity of plants and animals, which has been associated with an increased prevalence of allergic disease. The biodiversity hypothesis states that contact with natural environments enriches the human microbiome, promotes immune balance, and protects from allergy and inflammatory disorders [24]. Based on the findings of this longitudinal study, we postulated that the effects of environmental factors which cause allergic diseases present temporally different from person to person and depend on the type of disease.

Modern high-rise buildings and many indoor facilities attempt to save costs by minimizing energy consumption. Such attempts, however, could be the cause of increasing concentrations of indoor pollutants. Recent mechanistic studies investigating the effects of pollutant exposure on the development of allergic disease have been guided by key findings from birth cohorts, including the role of oxidative stress on promoting epigenetic modifications that regulate the gene expression of Treg cells through microRNAs (miRNAs) and DNA methylation [25,26].

This study had several strengths. Firstly, we used the ISAAC questionnaire, a method which has been used and validated in a number of Korean studies. Secondly, the disease prevalence data were based on physician-diagnosed and -treated disease, reducing the bias typically occurring in cross-sectional studies. A limitation of this study, however, is that, since the potential risk factors for each allergic disease are influenced by the changes in social and environmental conditions, the exact impact of environmental factors on allergic disease could not be analyzed, as not all other external factors can be controlled.

## 5. Conclusions

Allergic diseases result from an interaction between individual genetic susceptibility and exposure to environmental factors. This longitudinal study in Korea used The International Study of Asthma and Allergies in Childhood questionnaire and generalized estimating equations to analyze the influence of various environmental factors on the prevalence and course of allergic diseases, such as allergic rhinitis, atopic dermatitis, and allergic conjunctivitis. The authors found that (1) allergic rhinitis and atopic dermatitis were associated with indoor pollution, (2) allergic conjunctivitis was associated with indoor chemical compounds and low levels of weekly physical activity. This study suggested that the proper evaluation and decrease in the number of environmental risk factors could effectively manage allergic diseases.

This study analyzed the impact of indoor environmental factors and physical activity on the prevalence changes in allergic diseases in elementary school students in Ulsan, Korea. However, some questions could be further studied, such as the relationship between the surrounding climate, airborne allergens, food allergens, or emotional factors and atopic diseases.

## Figures and Tables

**Table 1 ijerph-17-08831-t001:** General characteristics of the subjects at the first survey (*n* = 390).

Variables	Classification	Frequency (%)
Sex	Male	195 (50.0)
	Female	195 (50.0)
Age (year)		8.4 ± 0.5
Follow-up period (Months)		49.0 ± 4.7
Prevalence		
Allergic rhinitis	Physician diagnosed	88 (22.6)
	Physician treated	104 (26.7)
Atopic dermatitis	Physician diagnosed	37 (9.5)
	Physician treated	31 (7.9)
Allergic conjunctivitis	Physician diagnosed	37 (9.5)
	Physician treated	33 (8.5)
Family history		
Allergic rhinitis	Maternal	86 (22.3)
	Siblings	46 (12.2)
Atopic dermatitis	Maternal	14 (3.7)
	Siblings	40 (10.6)
Allergic conjunctivitis	Maternal	28 (7.4)
	Siblings	41 (10.9)

**Table 2 ijerph-17-08831-t002:** Comparison of individual changes in the prevalence of allergic diseases during the first and second survey.

Allergicdisease	Diagnosed by Physician			*p* *	Treated by Physician			*p* *
	Second	First			Second	First		
		No	Yes			No	Yes	
AR	No	238 (63.3)	37 (9.8)	0.198	No	218 (58.8)	47 (12.7)	0.919
	Yes	50 (13.3)	51 (13.6)		Yes	49 (13.2)	57 (15.4)	
AD	No	326 (85.6)	26 (6.8)	0.291	No	327 (87.9)	24 (6.5)	0.143
	Yes	18 (4.7)	11 (2.9)		Yes	14 (3.8)	7 (1.9)	
AC	No	304 (80.4)	24 (6.3)	0.124	No	303 (81.0)	23 (6.1)	0.072
	Yes	37 (9.8)	13 (3.4)		Yes	38 (10.2)	10 (2.7)	

AR: allergic rhinitis; AD: atopic dermatitis; AC: allergic conjunctivitis. *p* *: calculated by McNemar test.

**Table 3 ijerph-17-08831-t003:** Comparison of changes in the individual risk of environmental factors between the first and second survey (continuous variables).

Variables	Time of Survey			*p* *
	First	Second	Difference	
House income (rank)	3.61 ± 0.96	4.59 ± 1.29	0.97 ± 0.97	<0.001
Time spent of television and computer (hours)	1.43 ± 0.59	1.68 ± 0.71	0.26 ± 0.69	<0.001
Moderate physical activity per week	3.42 ± 1.87	3.16 ± 1.84	−0.26 ± 2.28	0.025
Time spent at home (hours)	13.48 ± 4.47	12.24 ± 3.22	−1.13 ± 5.06	<0.001
Time spent at school (hours)	4.67 ± 0.97	6.62 ± 1.66	1.94 ± 1.87	<0.001
Daily ventilation time (hours)	4.26 ± 4.98	2.1 ± 2.98	−2.02 ± 5.25	<0.001
Current building use (years)	3.58 ± 0.86	3.67 ± 0.71	0.09 ± 0.9	0.042

*p* *: calculated by paired *t*-test.

**Table 4 ijerph-17-08831-t004:** Comparison of changes in the individual risk of environmental factors between the first and second survey (categorical variables).

Variables	Environmental Risk Factors			*p* *
	Second	First		
		No	Yes	
Paternal smoking	No	194 (49.7)	0 (0.0)	<0.001
	Yes	20 (5.1)	176 (45.1)	
Maternal smoking	No	380 (97.4)	0 (0)	0.063
	Yes	5 (1.3)	5 (1.3)	
Secondhand smoking	No	266 (70.2)	41 (10.8)	0.188
	Yes	29 (7.7)	43 (11.3)	
Remodeling of room	No	298 (81.2)	42 (11.4)	0.025
	Yes	23 (6.3)	4 (1.1)	
Painting of internal wall	No	299 (82.1)	37 (10.2)	0.162
	Yes	25 (6.9)	3 (0.8)	
Replacement of wallpaper and floor	No	249 (66.8)	67 (18.0)	0.036
	Yes	44 (11.8)	13 (3.5)	
Use of sleeping bed	No	80 (21.2)	35 (9.3)	0.001
	Yes	68 (45.95)	194 (51.5)	
Use of air cleaner	No	233 (63.7)	44 (12.0)	0.661
	Yes	39 (10.7)	50 (13.7)	
Use of humidifier	No	225 (64.2)	74 (18.6)	<0.001
	Yes	27 (6.8)	41 (10.3)	
Use of air conditioner	No	42 (11.0)	12 (3.1)	<0.001
	Yes	37 (9.7)	291 (76.2)	
Use of aromatics	No	250 (67.4)	37 (10.0)	0.326
	Yes	47 (12.7)	37 (10.0)	
Use of insecticide	No	259 (71.2)	59 (16.2)	<0.001
	Yes	15 (4.1)	31 (8.5)	
Use of hair spray	No	310 (85.2)	25 (6.9)	0.154
	Yes	15 (4.1)	14 (3.8)	
Movement or repair of new house	No	172 (44.8)	68 (17.7)	0.007
	Yes	39 (10.2)	105 (27.3)	
Pet ownership	No	242 (63.7)	26 (6.8)	<0.001
	Yes	67 (17.6)	45 (11.8)	
Irritation symptoms of air pollution	No	284 (76.1)	33 (8.8)	1.000
	Yes	34 (9.1)	22 (5.9)	

*p* *: calculated by McNemar test.

**Table 5 ijerph-17-08831-t005:** Results of generalized estimating equations for allergic disease prevalence changes as a function of environmental risk factors.

Allergic Disease	Diagnosed by Physician			Treated by Physician		
Variables	OR	95% CI	*p*	OR	95% CI	*p*
Allergic Rhinitis						
Family history (mother)	4.129	2.401–7.101	<0.001	2.792	1.660–4.696	<0.001
Secondhand smoking	1.722	1.030–2.878	0.038	1.740	1.073–2.821	0.025
Remodeling of room	1.033	0.361–2.958	0.951	7.437	1.761–31.413	0.006
Replacement of wallpaper and floor	0.951	0.503–1.796	0.876	1.820	1.032–3.209	0.039
Use of humidifier	1.935	1.207–3.100	0.006	1.661	1.055–2.617	0.029
Irritation for air pollution	1.797	1.019–3.167	0.043	1.537	0.886–2.668	0.126
Atopic Dermatitis						
Family history (mother)	5.262	1.714–16.155	0.004	4.652	1.405–15.403	0.012
Family history (sibling)	0.960	0.932–0.989	0.007	0.971	0.946–0.997	0.029
Time spent at school	0.772	0.610–0.978	0.032	0.983	0.733–1.318	0.910
Movement or repair of new house	2.702	1.412–5.171	0.003	2.264	1.070–4.792	0.033
Irritation for air pollution	0.412	0.158–1.075	0.070	1.181	0.425–3.281	0.749
Allergic Conjunctivitis						
Family history(mother)	2.763	1.385–5.511	0.004	3.342	1.707–6.544	<0.001
House income (rank)	1.583	1.147–2.187	0.005	1.465	1.058–2.027	0.021
Moderate physical activity per week	1.169	1.005–1.359	0.042	1.151	0.98–1.352	0.087
Painting of internal wall	0.200	0.056–0.714	0.013	0.129	0.021–0.778	0.025
Use of insecticide	2.414	1.310–4.448	0.005	2.415	1.263–4.616	0.008
Use of hair spray	0.338	0.123–0.930	0.036	0.460	0.163–1.297	0.142
Irritation for air pollution	2.207	1.041–4.679	0.039	2.295	1.056–4.985	0.036

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
