# Peer review of "Impact of Environmental Factors on the Prevalence Changes of Allergic Diseases in Elementary School Students in Ulsan, Korea: A Longitudinal Study"

_ijerph, 2020, doi:10.3390/ijerph17238831_

Round 1

Reviewer 1 Report

Title

  • Suggest adding the country the study was done to the title to give the readers a heads-up on the study location. 

Abstract

  • lack of specific details especially on the main findings. 

Introduction

  • while the study itself is good, the introduction is too brief to convince the readers on the importance of the research and the research gap. 

Methods

  • section 2.2 - include software used for the analysis

Results

  • avoid repeating information already available in the tables in the text. Use the text to highlight only important findings from the tables
  • what will be the significance of table 3.1 and 3.2 if GEE ( table 4) is eventually used to provide more accurate changes between the time points? 

Discussion

  • I have no additional comments here

Do consider expanding the conclusion. 

Author Response

Reviewer 1

Comments and Suggestions for Authors

Title

Suggest adding the country the study was done to the title to give the readers a heads-up on the study location.

Abstract

lack of specific details especially on the main findings.

Answer: we changed the last sentence of abstract as followings

“To effectively manage allergic diseases, proper evaluation of how environmental factors affect temporally on each allergic disease should be preceded, followed by appropriate interventions.”

Introduction

while the study itself is good, the introduction is too brief to convince the readers on the importance of the research and the research gap.

Answer: according to reviewer’s comments, we added some explanations for relationship between allergic diseases and the related factors in introduction as followings; P3, L49-55

The factors associated with prevalence of allergic diseases and comorbidity were varied. The climate related factors including ambient temperature and outdoor humidity would be associated with allergic diseases symptoms. Dietary factors and socioeconomic status may also affect the prevalence directly and indirectly. Air pollution not only aggravates symptoms of allergic diseases, but also causes onsets in new individuals. In specific, the oxidizing agents including nitrogen oxides and ozone are well recognized as important allergy associated pollutants.

Methods

section 2.2 - include software used for the analysis

Answer: we included the information of software in manuscripts as followings: “Statistical analysis were carried out by Statistical analysis Software (SAS, 9.4 SAS Institute Inc. Cary, North Carolina, USA).”

Results

avoid repeating information already available in the tables in the text. Use the text to highlight only important findings from the tables

Answer: We try to avoid repeating information in table and text. And described it briefly.

what will be the significance of table 3.1 and 3.2 if GEE ( table 4) is eventually used to provide more accurate changes between the time points?

Answer: The table 3.1 and 3.2 were a kind of univariate analysis results of two time points for the environmental factors, which we surveyed. GEE shows adjusted results of significant factors for prevalence changes. Therefore, both of them are necessary.

Discussion

I have no additional comments here

Do consider expanding the conclusion.

Answer: we changed the following sentence in conclusion. “Allergic diseases result from an interaction between individual genetic susceptibility and exposure to environmental factors. This longitudinal study in Korea used The International Study of Asthma and Allergies in Childhood questionnaire and generalized estimating equations to analyze that various environmental factors influence prevalence and course of allergic diseases, like allergic rhinitis, atopic dermatitis and allergic conjunctivitis. The authors found that 1) allergic rhinitis and atopic dermatitis were associated with indoor pollution, 2) allergic conjunctivitis was associated with indoor chemical compounds and low levels of weekly physical activity. This study suggested that proper evaluation and the decreasing of environmental risk factors could effectively manage allergic diseases.

This study analyzed that impact of indoor environmental factors and physical activity on the prevalence changes of allergic diseases in elementary school students in Ulsan, Korea. However, some questions could be further studied, such as, the relationship between surrounding climate, airborne allergens, food allergens or emotional factors and atopic diseases.”

Thank you for the thoughtful comments. The manuscript has improved a lot than before.

Reviewer 2 Report

Allergic diseases result from an interaction between individual genetic susceptibility and exposure to environmental factors. This longitudinal study in Korea used The International Study of Asthma and Allergies in Childhood questionnaire and generalized estimating equations to analyze that various environmental factors influence prevalence and course of allergic diseases, like allergic rhinitis, atopic dermatitis and allergic conjunctivitis.

The authors found that 1) allergic rhinitis and atopic dermatitis were associated with indoor pollution, 2) allergic conjunctivitis was associated with indoor chemical compounds and low levels of weekly physical activity. This study suggested that proper evaluation and the decreasing of environmental risk factors could effectively manage allergic diseases.

This study analyzed that impact of indoor environmental factors and physical activity on the prevalence changes of allergic diseases in elementary school students in Ulsan, Korea. However, some questions could be further studied, such as, the relationship between surrounding climate, airborne allergens, food allergens or emotional factors and atopic diseases.

Author Response

Reviewer 2

Comments and Suggestions for Authors

Allergic diseases result from an interaction between individual genetic susceptibility and exposure to environmental factors. This longitudinal study in Korea used The International Study of Asthma and Allergies in Childhood questionnaire and generalized estimating equations to analyze that various environmental factors influence prevalence and course of allergic diseases, like allergic rhinitis, atopic dermatitis and allergic conjunctivitis.

The authors found that 1) allergic rhinitis and atopic dermatitis were associated with indoor pollution, 2) allergic conjunctivitis was associated with indoor chemical compounds and low levels of weekly physical activity. This study suggested that proper evaluation and the decreasing of environmental risk factors could effectively manage allergic diseases.

This study analyzed that impact of indoor environmental factors and physical activity on the prevalence changes of allergic diseases in elementary school students in Ulsan, Korea. However, some questions could be further studied, such as, the relationship between surrounding climate, airborne allergens, food allergens or emotional factors and atopic diseases.

Answer: We changed the sentence in conclusion according to reviewer’s comments.

Thank you for the precious and thoughtful comments. The manuscript has improved a lot than before.

Reviewer 3 Report

The subject of this study is very important, and the findings contribute to increase the knowledge in this field involving the impact of environmental factors on the prevalence changes of allergic diseases in students.

 Lines 21-22 – It is not necessary to use the acronym ISAAC in …(2013–2014). The International Study of Asthma and Allergies in Childhood (ISAAC) questionnaire…

lines 33-34 - I suggest to improve the conclusion

Line 44- It would be important to clarify the country in “In Korea,…and … in all the manuscript

Line 59 – I suggest to add the hypothesis of the study at the end of the Introduction section

Line 66 – to define ISAAC in the first time in the text …The standardized ISAAC

line 69 - to clariy about the additional questionnaire..An additional questionnaire...is it a validated questionnaire?

Line 169 – I suggest to start the Discussion considering the hypothesis…for example.. As it was hypothesized..

line 252 - I suggest to improve the conclusion

I am suggesting below some references that the authors could add to the Introduction and/or Discussion section

1: Alkotob SS, Cannedy C, Harter K, Movassagh H, Paudel B, Prunicki M, Sampath

V, Schikowski T, Smith E, Zhao Q, Traidl-Hoffmann C, Nadeau KC. Advances and

novel developments in environmental influences on the development of atopic

diseases. Allergy. 2020 Oct 9. doi: 10.1111/all.14624. Epub ahead of print.

PMID: 33037680.

2: Lin X, Ren X, Xiao X, Yang Z, Yao S, Wong GW, Liu Z, Wang C, Su Z, Li J.

Important Role of Immunological Responses to Environmental Exposure in the

Development of Allergic Asthma. Allergy Asthma Immunol Res. 2020

Nov;12(6):934-948. doi: 10.4168/aair.2020.12.6.934. PMID: 32935487; PMCID:

PMC7492518.

3: Krishna MT, Mahesh PA, Vedanthan P, Moitra S, Mehta V, Christopher DJ. An

appraisal of allergic disorders in India and an urgent call for action. World

Allergy Organ J. 2020 Aug 1;13(7):100446. doi: 10.1016/j.waojou.2020.100446.

PMID: 32774662; PMCID: PMC7398972.

4: Miyazaki D, Fukagawa K, Okamoto S, Fukushima A, Uchio E, Ebihara N, Shoji J,

Namba K, Shimizu Y. Epidemiological aspects of allergic conjunctivitis. Allergol

Int. 2020 Oct;69(4):487-495. doi: 10.1016/j.alit.2020.06.004. Epub 2020 Jul 9.

PMID: 32654975.

5: Barnthouse M, Jones BL. The Impact of Environmental Chronic and Toxic Stress

on Asthma. Clin Rev Allergy Immunol. 2019 Dec;57(3):427-438. doi:

10.1007/s12016-019-08736-x. PMID: 31079340.

Author Response

Reviewer 3

Comments and Suggestions for Authors

The subject of this study is very important, and the findings contribute to increase the knowledge in this field involving the impact of environmental factors on the prevalence changes of allergic diseases in students.

Lines 21-22 – It is not necessary to use the acronym ISAAC in …(2013–2014). The International Study of Asthma and Allergies in Childhood (ISAAC) questionnaire…

Answer: we deleted the acronym ISAAC in abstract.

lines 33-34 - I suggest to improve the conclusion

Answer: we changed the conclusion in abstract, also added the following sentence in conclusion.

P2 L 32-34 “To effectively manage allergic diseases, proper evaluation of how environmental factors affect temporally on each allergic disease should be preceded, followed by appropriate interventions.”

P10 L271-272 “Further studies will be necessary for the relationship between urbanization, surrounding climate, airborne allergens, food allergens or emotional factors and allergic diseases.”

Line 44- It would be important to clarify the country in “In Korea,…and … in all the manuscript

Answer: To clarify the country, we referred the country in the title as followings: “Impact of environmental factors on the prevalence changes of allergic diseases in elementary school students in Ulsan, Korea: A longitudinal study”

Line 59 – I suggest to add the hypothesis of the study at the end of the Introduction section

Answer: The hypothesis of the study was given as the study aim at the end of the Introduction section as followings: “to evaluate the impact of environmental risk factors on the changes in the prevalence of different allergic diseases”.

Line 66 – to define ISAAC in the first time in the text …The standardized ISAAC

International Study of Asthma and Allergies in Childhood (ISAAC)

Answer: we defined International Study of Asthma and Allergies in Childhood (ISAAC) in the manuscript according to comments.

line 69 - to clariy about the additional questionnaire. An additional questionnaire...is it a validated questionnaire?

Answer: The additional questionnaire have used in this cohort study from 2009. This additional questionnaire were widely used for the research tool after defining through the process of discussion from 5 environmental health center in Korea. Therefore, we thought that the questionnaire was validated already.

Line 169 – I suggest to start the Discussion considering the hypothesis…for example.. As it was hypothesized..

Answer: We did seriously consider reviewer’s suggestion. As we mentioned before, to avoid repetition, the aim of this study could be substituted the hypothesis.

line 252 - I suggest to improve the conclusion

Answer: we changed conclusion as followings. P10 L266-278 “Allergic diseases result from an interaction between individual genetic susceptibility and exposure to environmental factors. This longitudinal study in Korea used The International Study of Asthma and Allergies in Childhood questionnaire and generalized estimating equations to analyze that various environmental factors influence prevalence and course of allergic diseases, like allergic rhinitis, atopic dermatitis and allergic conjunctivitis. The authors found that 1) allergic rhinitis and atopic dermatitis were associated with indoor pollution, 2) allergic conjunctivitis was associated with indoor chemical compounds and low levels of weekly physical activity. This study suggested that proper evaluation and the decreasing of environmental risk factors could effectively manage allergic diseases. This study analyzed that impact of indoor environmental factors and physical activity on the prevalence changes of allergic diseases in elementary school students in Ulsan, Korea. However, some questions could be further studied, such as, the relationship between surrounding climate, airborne allergens, food allergens or emotional factors and atopic diseases.”

I am suggesting below some references that the authors could add to the Introduction and/or Discussion section

Answer: According to reviewer’s suggestion, we added to Introduction and Discussion section as followings:

In introduction section P3 L48-53

The factors associated with prevalence of allergic diseases and comorbidity were varied. The climate related factors including ambient temperature and outdoor humidity would be associated with allergic diseases symptoms. Dietary factors and socioeconomic status may also affect the prevalence directly and indirectly. Air pollution not only aggravates symptoms of allergic diseases, but also causes onsets in new individuals. In specific, the oxidizing agents including nitrogen oxides and ozone are well recognized as important allergy associated pollutants.

In discussion section P3 L48-53

Also, the hygiene hypothesis conceptualized that childhood exposure to microorganisms will protect also against allergy.3 The exposome, which is the sum total of all the exposures of an individual in a lifetime, has undergone major shifts in the last few decades, affecting human health and disease. Urbanization has led to reduced biodiversity of plants and animals, which has been associated with increased allergic disease. The biodiversity hypothesis states that contact with natural environments enriches the human microbiome, promotes immune balance and protects from allergy and inflammatory disorders.

Thank you for the thoughtful comments. The manuscript has improved a lot than before.
